# Insight into the Fecal Microbiota Signature Associated with Growth Specificity in Korean Jindo Dogs Using 16S rRNA Sequencing

**DOI:** 10.3390/ani12192499

**Published:** 2022-09-20

**Authors:** So-Young Choi, Bong-Hwan Choi, Ji-Hye Cha, Yeong-Jo Lim, Sunirmal Sheet, Min-Ji Song, Min-Jeong Ko, Na-Yeon Kim, Jong-Seok Kim, Seung-Jin Lee, Seok-Il Oh, Won-Cheoul Park

**Affiliations:** 1Animal Genome and Bioinformatics, National Institute of Animal Science, Rural Development Administration, Iseo-myeon, Wanju-gun 55365, Korea; 2Animal Genetic Resources Research Center, National Institute of Animal Science, Rural Development Administration, Hamyang 50000, Korea; 3Korean Jindo and Domestic Animals Center, Jindo-gun 58915, Korea

**Keywords:** Jindo-dog, feces, microbiome, age

## Abstract

**Simple Summary:**

Microbiomes have a proven effect on canine growth; however, there is a lack of research on the microbiomes associated with growth and development in Jindo dogs. The results of this study showed compositional differences in the fecal microbiome between young and adult dogs. We confirmed the composition and functional differences in the fecal microbial community genes between 4- and 16-month-old dogs in this study. To the best of our knowledge, this is the first report of the fecal microbiome based on the growth stages of Jindo dogs.

**Abstract:**

Gut microbiomes are well recognized to serve a variety of roles in health and disease, even though their functions are not yet completely understood. Previous studies have demonstrated that the microbiomes of juvenile and adult dogs have significantly different compositions and characteristics. However, there is still a scarcity of basic microbiome research in dogs. In this study, we aimed to advance our understanding by confirming the difference in fecal microbiome between young and adult dogs by analyzing the feces of 4-month and 16-month-old Jindo dogs, a domestic Korean breed. Microbiome data were generated and examined for the two age groups using 16S rRNA analysis. Comparison results revealed that the 16-month-old group presented a relatively high distribution of *Bacteroides*, whereas the 4-month-old group presented a comparatively high distribution of the *Lactobacillus* genus. Microbial function prediction analyses confirmed the relative abundance of lipid metabolism in 4-month-old dogs. In 16-month-old dogs, glucose metabolism was determined using microbial function prediction analyses. This implies that the functional microbiome changes similarly to the latter in adults compared with childhood. Overall, we discovered compositional and functional variations between genes of the gut microbial population in juveniles and adults. These microbial community profiles can be used as references for future research on the microbiome associated with health and development in the canine population.

## 1. Introduction

According to research by the American Pet Products Association, 69 million out of 90.5 million households have dogs [1]. In Korea, the pet industry is expected to grow to more than 4.7 billion dollars by 2027, with the number of households estimated to raise 6.04 million pets [2,3]. Among these, Jindo dogs are representative of indigenous dog breeds in Korea. In total, 19,000 Jindo dogs have been reported in South Korea [4]. The life expectancy of dogs has increased; however, the basic characteristics of healthy dogs are unknown, warranting research. Therefore, researchers are studying many ways to raise healthy dogs, and studying the microbiome is one of the themes. A microbiome is a community of microorganisms in the environment and its entire set of genes. The main field studied in animals in various environments is intestinal microorganisms, and it is known that intestinal microorganisms are related to host health conditions such as immunity, metabolic diseases, and neurological organs [5,6,7].

Many studies have shown that microbiomes not only exist in the environment but also influence each other and the host symbiotically. Microbiota signatures vary according to the environment, diet, and different stages of growth [8,9]. An animal’s growth, determined by its weight and overall size, depends on three major factors: genetic pool, nutrient supply, and the environment [10]. In recent years, intestinal microorganisms have been reported to be the main intermediaries connecting hosts, nutrients, and the environment. Therefore, it is essential to identify microbial changes and their functions according to the host growth stages [11]. Most microbiome research focuses on adults; however, the composition of microbiomes changes significantly throughout a lifetime [12,13]. In humans, intestinal microorganisms are known to affect adults from an early age, and those established in infancy and childhood are related to diseases such as obesity, asthma, and inflammatory bowel disease [14,15,16].

In addition, healthy intestinal microbiomes can regulate bioavailability, which can inhibit the entry of pathogenic bacteria and regulate growth hormones. Therefore, microbiomes in growing dogs are very important for healthy growth and development [9]. Some canine studies on aging have reported an age-appropriate microbiome, where the older the dog, the lower the microbial [17] and *Fusobacteria* abundance [18]. However, research on the dog microbiome is at a nascent stage, and there is a lack of research on the microbial characteristics of growing dogs. Therefore, we analyzed the fecal microbiota composition in Jindo dogs, which have been rarely studied, in 4-month-old and 16-month-old dogs to elucidate the changes in age-dependent microorganisms in the same dogs to identify potentially beneficial microorganisms for increasing the dog’s health status.

## 2. Materials and Methods

### 2.1. Animals and Sample Collection

The animal procedures performed in this study were reviewed and approved by the Institutional Animal Care and Use Committee of the National Institute of Animal Science (NIAS), Korea (approval number: NIAS-2020-467). Six Jindo dogs (three males and three females), born in January 2020, were used for the study. They were housed individually and raised in only one commercial Jindo-dog breeding facility in Jindo-gun. Korean Jindo and Domestic Animals Center veterinarians monitored the dogs. After weaning, they were fed the same feed during the experiment (vita-dog pride, DONGAONE). Fecal samples were collected between 7–9 a.m. when dogs were 4 and 16 months old. The collected feces were stored at −20 °C immediately before microbial DNA extraction. The dogs had no history of disease or castration.

### 2.2. Extracting Microbial DNA and 16S rRNA Sequencing

Microbial DNA was extracted from fecal samples using the ReliaPrep gDNATissue Miniprep System (PROMEGA Inc., Madison, WI, USA) according to the manufacturer’s protocol. The extracted DNA was stored at −20 °C until sequencing data were produced.

DNA samples were amplified by the hypervariable regions V3–V4 specific primers (adaptor/sequencing primer/specific locus primer: 341F: 5′-CCTACGGGNGGCWGCAG-3′, 806R: reverse 5′-ACTACHVGGGTATCTAATCC-3′) of the 16S rRNA gene sequence in bacteria. The PCR amplification was performed under thermal cycling conditions. The DNA was subjected to an initial denaturation at 95 °C for 3 min, followed by 25 cycles of denaturation at 95 °C for 30 s, and a final elongation step at 72 °C for 5 min. The amplicon products were purified using AMPure XP beads (Beckman Coulter, Nyon, Switzerland). Secondary amplification was conducted using the first PCR amplicon products and attaching an adaptor under the first amplification conditions, but only for eight cycles. DNA quality and product size were evaluated on a Bioanalyzer 2100 (Agilent, Palo Alto, CA, USA) using a DNA 7500 chip. The sequencing was performed on the Illumina MiSeq platform with 2 × 300 bp paired-end reads (Illumina, Inc., San Diego, CA, USA).

### 2.3. Bioinformatics Analysis

Demultiplexed sequences were processed using Quantitative Insight into Microbial Ecology 2 (QIIME2, 2021.4) [19] using default parameters. The sequence data were converted into QIIME2 artifacts. DADA2 was used to remove non-biological nucleotides, quality filtering and denoising were performed, and low-quality sequences were removed with a quality score (<Q 25) and amplicon sequence variant calling (ASVs) using the qiime dada2-denoise-paired method [20]. ASV feature counts of the 16S rRNA sequence were classified using a pre-trained naive Bayes classifier, which was trained on the SILVA 138 SSU database used for the assigned taxonomy ID on the alignment with the classification of different levels [21]. The Shannon diversity index and richness (number of observed features) were calculated using QIIME2 to evaluate alpha diversity. Beta diversity was presented using principal coordinate analysis (PCoA) and estimated by the distance between the microbial compositions of samples calculated using Bray–Curtis dissimilarity. Significant differences in microbial composition between the two groups were assessed by the permutational multivariate analysis of variance (PERMANOVA). Visualization of diversity was performed using the phyloseq package in R software [22].

### 2.4. LEfSe and Co-Abundance Network Analysis

The linear discriminant analysis (LDA) effect size (LEfSe) method was conducted to detect any bacterial taxon having a significantly different abundance between the 4-month and 16-month age groups, and visualization was performed using the web-based tool Galaxy (http://huttenhower.sph.harvard.edu/galaxy/ (accessed on 25 June 2022) for LEfSe analysis. Statistically significant taxa were reported with linear discriminant analysis (LDA) scores > 4 [23].

The correlation of the genera for each age group was statistically analyzed using the Pearson correlation. We used Cytoscape 3.9.1 [24] to visualize the network and identify co-abundant genera. All the used genera nodes were R > |0.8|.

### 2.5. Functional Prediction of Microbial Gene

The microbial functions were predicted based on the identified taxonomic ASVs. The abundance of microbial gene function was assigned using phylogenetic investment by reconstruction of unobserved States2 (PICRUSt2) [25]. A comparison of the measured predictive gene clustered ortholog genes (COGs) between age groups was performed using ALDEx2 [26] in the R package. The ALDEx2 algorithm was used to calculate the effect size of the COGs and identify significantly different COGs between the growth stages. The COGs result was a functional category assignment using STAMP software [27]. Clustered ortholog genes were used to identify distinct COGs between the age groups with effect size > |3| and Benjamin–Hochberg (BH) adjusted *p*-value < 0.05 (Wilcoxon rank-sum test), and the results were presented using an MA plot. Subsequently, significantly different COGs were expressed as a heatmap (R package, pheatmap) [28]. These COGs represent hierarchical annotations of orthologous groups. At the next higher level, the COGs were grouped into pathways, and at the third and highest levels, the pathways were grouped into four categories (including those poorly characterized). The hierarchical cladogram of the COGs was schematized using Galaxy [23].

## 3. Results

### 3.1. Sequencing Results Summary and Bacterial Structure

Fecal samples were collected from six Jindo dogs aged 4 and 16 months. Microbial DNA was extracted from 12 fecal samples, and the V3–V4 region of the 16S rRNA was sequenced. A total of 1,091,273 reads were generated, and 90,939 reads were produced, on average, per sample. After merging paired-end reads, the chimeric sequence was removed simultaneously with quality filtering (Q score < 25), and an average of 51,215 reads were obtained. Compared to the raw reads, approximately 54% of high-quality reads were obtained (Table 1 and Appendix A). Taxonomic classification was performed using the microorganism database (SILVA 138). Figure 1a,b shows the taxonomic analysis. At the phylum level, *Firmicutes*, *Actinobacteria*, *Bacteroidota*, *Proteobacteria*, and *Fusobacteria* accounted for >1.5% in both months. At the age of 4 months, *Firmicutes* were identified in 77% and *Bacteroidota* in 12% of samples. In the 16-month-old group, *Firmicutes* were identified in 45% and *Bacteroidota* in 43% (Figure 1a and Appendix A). At the genus level, *Lactobacillus* (27.1%), *Muribaculaceae* (9%), *Blautia* (7%), *Allobaculum* (4.5%), and *Bifidobacterium* (4.3%) accounted for >1%. *Prevotella* (23.8%), *Bacteroides* (11%), *Fusobacterium* (5%), and *Megamonas* (4%) were identified in the 16-month-old group. (Figure 1b and Appendix A). Neither observed-ASVs nor the Shannon index could identify any significant differences in growth as an alpha-diversity analysis (Kruskal–Wallis, observed-ASVs *p*-value = 0.42 and Shannon = 0.2). Beta diversity is shown by the dissimilarity among samples, according to the microbial composition depicted in the PCoA analysis (Bray–Curtis). The microbial composition of the 4-month-old and 16-month-old groups was confirmed to be different (Figure 1d).

### 3.2. LEfSe Analysis and Co-Occurrence Analysis among the Microbial Genera

A LEfSe analysis was performed to identify microorganisms that differed between the 4-month-old and 16-month-old groups. A total of 23 taxa with significant differences were identified. In the 4-month-old group, the abundance of 10 microorganisms was significantly higher than that of the 16-month-old group (*c_Baciili*, *p_Frimicutes*, *g__Lactobacillus*, *f_Lactobacilliaeceae*, *o_Lactobacillales*, *o_Erysipelotrichales*, *f_Erysipelotrichaceae*, *p_Actinobacteriota*, *c_Bacilli*, *g_uncultured*, and *g_Blautia*), and in the 16-month-old group, 13 microorganisms were significantly higher than that for the 4-month-old group (*o_Bacteroidales*, *c_Bacteroidia*, *p_Bacteroidota*, *f_Prevotellaceae*, *g_Prevotella*, *g_Bacteroides*, *f_Bacteroidaceae*, *c_Negativicutes*, *o_Veillonellales_Selenomonadales*, *o_Oscillospirales*, *f_Ruminococcaceae*, *f_Selenomonadaceae*, and *g_Megamonas*). In the cladogram, *Actinobacteria* and *Firmicutes* were confirmed in the 4-month-old group. The 16-month-old group showed microorganisms belonging to the *Bacteroidota* and some *Firmicutes* phyla (Figure 2a,b, LDA > 4).

We investigated the connections between the different types of microorganisms based on their growth. A bacterial co-occurrence network was created using the Pearson correlation between the relative abundances of the genera. In the correlation analysis, 34 genera were found in the 4-month-old group. Consequently, 29 genera were correlated. *Lactobacillus*, which had the largest distribution in the 4-month-old group, was found to have 13 different types of microbes with a significant correlation. *Lactobacillus* was negatively correlated with *f_Erysipelotrichaceae*, *g_uncultured*, *f_Lachnospiraceae_*, *g_Ruminococcus._gnavus_group*, *g_.Ruminococcus._torques_group,* and *g_Fusobacterium*. Furthermore, there was a positive correlation between *g_Holdemanella*, *g_Peptoclostridium*, *g_Romboutsia*, *g_Clostridium_sensu_stricto_1*, *g_Erysipelatoclostridium*, and *g_Escherichia.Shigella* (Figure 2c and Appendix A). In the 16-month-old group, 45 genera were identified in all samples and analyzed. Consequently, 38 genera were found to be associated. *Prevotella* showed the highest distribution and was negatively correlated with *Fournierella*. *Bacteroides*, which showed the next highest distribution, were positively correlated with eight genera, *g_Fusobacterium*, *g_Intestinimonas*, *g_Lachnospiraceae_NK4A136_group*, *g_Phascolarctobacterium*, *g_Prevotellaceae_Ga6A1_group*, *g_Sutterella*, *f_Erysipelotrichaceae_g_uncultured*, and *g_Alloprevotella*. In addition, it was confirmed that *f_Erysipelotrichaceae_g_uncultured* and *g_Fusobacterium*, which showed a negative correlation with *Lactobacillus* at 4 months, were positively correlated with *Bacteroides* at 16 months (Figure 2d and Appendix A).

### 3.3. Microbial Functional Prediction and Distinction between Ages

PICRUSt2 was used to perform COGs to analyze the functional gene prediction of the microbiome in the 4- and 16-month-old groups. In total, 4270 COGs were identified (Appendix A). Using the ALDEx2 package in R, COGs distinguished between 4 and 16 months were identified. Significantly differentiated COGs between the two groups are shown using the MA plot (Figure 3a, Appendix A, effect size > |3|, adjusted *p*-value < 0.05). The functional categories assigned to 48 COGs, including the unknown function, were classified using the pheatmap package in R to show the difference between the 4-month-old and 16-month-old groups (Figure 3b). The predicted major COGs functions were categorized using a circular dendrogram (Figure 3c). Function-unknown COGs are excluded from the dendrogram. Major COGs classes were identified as “Metabolism”, “Information storage and processing”, and “Cellular processes and signaling”. In particular, the difference according to the category was confirmed in metabolism in the same class. The “Metabolism” of COGs classes, the categories of “G, Carbohydrate transport and metabolism”, “E, amino acid transport and metabolism”, and “F, Nucleotide transport and metabolism” is shown in the 16-month-old groups, and the “H, Coenzyme transport and metabolism” and “I, Lipid transport and metabolism” is shown in the 4-month-old group.

## 4. Discussion

This study aimed to determine changes in the fecal microbiome of Jindo dogs over time during different growth stages. Further, we attempted to eliminate the variations of dietary effects due to the intestinal microbiome by subjecting the Jindo dogs to the same feed throughout the experiment. In this study, we compared 4- and 16-month-old Jindo dogs representing puppies and adults; however, we did not analyze other age groups and suggest only information about the fecal microbiomes between these two age groups.

Prior studies identified the fecal microbiota composition according to the growth stage of dogs [17,29,30]. *Firmicutes*, *Fusobacteria*, *Bacteroides*, *Proteobacteria*, and *Actinobacteria* were common at the phylum level, whereas *Lactobacillus*, *Fusobacterium*, *Prevotella*, and *Bacteroides* were common at the genus level. These results were identical to those of our study, except for the dog environment and age, and it was determined that the dominant genera were consistent between the studies. Following this, changes in microbial species richness were investigated, but there were no significant results (Figure 1c). However, contrary to our findings, other studies using a similar experimental design showed changes in microbial species diversity. A significant difference was found in microbiome abundance and diversity between dogs before and after weaning and among puppies, adult dogs, and old dogs [17,31,32]. However, it has been observed that in humans, the microbiome of toddlers after dietary intake resembles that of adults. It is affected by diet, environmental elements, and immunological capacity formation [33,34]. The gut microbiome of dogs from one week to one year and the microbial community from 9 weeks of age revealed composition within the distribution range of adult dog microbial communities [35]. The number of microorganisms in pigs increased dramatically from birth to around 20 weeks of age and then fluctuated slightly at later ages, with no significant variation [36].

Each growth stage, as depicted in Figure 1d, was grouped by checking the distance between samples on PCoA to determine the compositional difference of microorganisms. The composition of the microbiome varies between young and adult dogs [32,37]. *Firmicutes* were prevalent in young dogs but declined as they grew older, and *Bacteroides* were more prevalent [32]. Although our data revealed no significant differences between the 4-month-old and 16-month-old dogs, the microbial composition was found to be different. In this study, a total of 23 genera were substantially different between the two age groups. Microorganisms belonging to *Bacteroidota* and some *Firmicutes* were detected in the 16-month-old group, and *Actinobacteria* and *Firmicutes* were identified in the 4-month-old group. (Figure 2). The beneficial bacteria *Lactobacillus* genus was considerably more abundant in the 4-month-old group. *Lactobacillus*, which aids in the digestion of lactose in breast milk, is prevalent in weaning infants [38]. However, two months post-weaning, the 4-month-old group was found to have higher *Lactobacillus* levels than the 16-month-old group in our study. *Bacteroidota* (phylum) were more prevalent in the 16-month-old group. *Bacteroides* (genus) are known to predominate in the animal gut and mostly utilize polysaccharides or simple sugars. Obese dogs on high-protein, low-carbohydrate diets have been shown to have fewer *Bacteroides* (phylum) and more *Firmicutes* (phylum) when they lose weight [39]. According to our study, as dogs mature, their carbohydrate intake increases with their food intake, increasing intestinal *Bacteroidota* (phylum).

*Lactobacillus*, which was found to be correlated with 13 genera in 4-month-old dogs, had the highest abundance (Figure 2). *Erysipelotrichaceae* and *Fusobacterium*, which were negatively correlated with *Lactobacillus*, were positively correlated with *Bacteroides*. At 16 months, *Bacteroides* were highly abundant and correlated with the other genera. In humans, *Fusobacterium* has been associated with periodontal diseases such as Lumiere syndrome [40]. Oral-derived *Fusobacterium* has also been shown to reach colon tumors via the circulatory system and promote colon tumor progression [41,42]. *Intestinimonas*, a strain correlated with *Bacteroides* in the 16-month-old group, has been examined for its ability to enable the digestion of indigestible polysaccharides (indigestible protein) and glucose metabolism [43]. *Holdemanella*, which exhibits a positive correlation with *Lactobacillus*, has been proven to aid obese mice in managing GLP-1 and reducing hyperglycemia. It also has anti-inflammatory and antitumor effects in the colon [44]. The relationship between microbial communities in the 4- and 16-month-old groups were identified, and it is speculated that the role of gut bacteria differs depending on age.

In the 4-month-old group, lipid metabolism was more enriched than in the 16-month-old group. Even if they ate the same diet, the consumed feed amount was different, and the microorganisms that decomposed the milk fat consumed as a child may have remained higher in the intestine than in 16-month-olds. *Lactobacillus*, significant in the 4-month-old group (Figure 2a), can create conjugated linoleic acid by utilizing the lipid metabolite, which improves bond formation and immunity [45,46]. *Holdemanella* is also involved in synthesizing unsaturated long-chain fatty acids (Figure 2c) [44]. In the 16-month-old dogs, carbohydrate metabolism was more enriched than in 4-month-old dogs. As adults consume more feed, they also consume more starch (carbohydrates), which leads to an increase in the number of microbial communities capable of breaking down these carbohydrates. We discovered that fecal microorganisms with polysaccharide-using genes were significant in 16-month-old dogs (Figure 2a,c). *Bacteroidota* uses carbohydrate-degrading enzymes to primarily break down polysaccharides. They decompose polysaccharides more directly than other phyla, *Firmicutes* and *Actinobacteria*, which show a large abundance [47]. However, changes in microbiome functionality between adults and juveniles, explained by the differences in biological metabolic activities, have yet to be demonstrated. After weaning, the gut microbiota of children is associated with their health. Allergic rhinitis is known to decrease intestinal microbial diversity during childhood. Having a low abundance of *Bifidobacterium* in children puts them at risk of becoming obese [14,48]. Additionally, the intestinal microbiome of newborns is related to immunity, which has an impact on the intestinal microbiome and immune system even after the child has grown up [49].

This study used 16S rRNA gene sequencing to examine the fecal microbiome of young and adult Jindo dogs for the first time to confirm the fecal microbiome composition according to growth. It was also confirmed by estimating the functional characteristics of the microorganisms predicted during their growth stage. However, our research focused on a single breed of Jindo dog, while previous investigations found that the microbiome varies by breed [50]. Future studies should include not only Jindo dogs, but also other commonly raised dog breeds. Furthermore, it is necessary to understand the fecal microbiome from lactation to adult dogs at the genomic level to elucidate the differences in the microbiome according to the growth stage. It is also necessary to analyze the metabolome with a metatranscriptome to perform functional analysis to improve dog health and growth.

## 5. Conclusions

The fecal microbiome analysis of Jindo dogs revealed a distinct difference in microbial composition between adults and juveniles. The 4-month-old dogs had a significantly higher level of *Lactobacillus* and showed that microorganisms have a function in lipid metabolism. However, the *Bacteroides* were significantly higher in the 16-month-old dogs, and their function in carbohydrate metabolism was confirmed. These results indicate that the composition and function of fecal microbiomes alter with aging. Additionally, our study showed that the degradation of nutrients may depend on the major microbial community. To the best of our knowledge, this is the first study on the fecal microbiome based on growth stages in Jindo dogs. Therefore, we hope that our research will be helpful in future studies on dogs and other pets. In further studies, the functional analysis of major microbial communities according to age is required, as well as the confirmation of changes in microbial communities from pre-weaning to the elderly and confirming connections with health and wellness.

## Figures and Tables

**Figure 1 animals-12-02499-f001:**
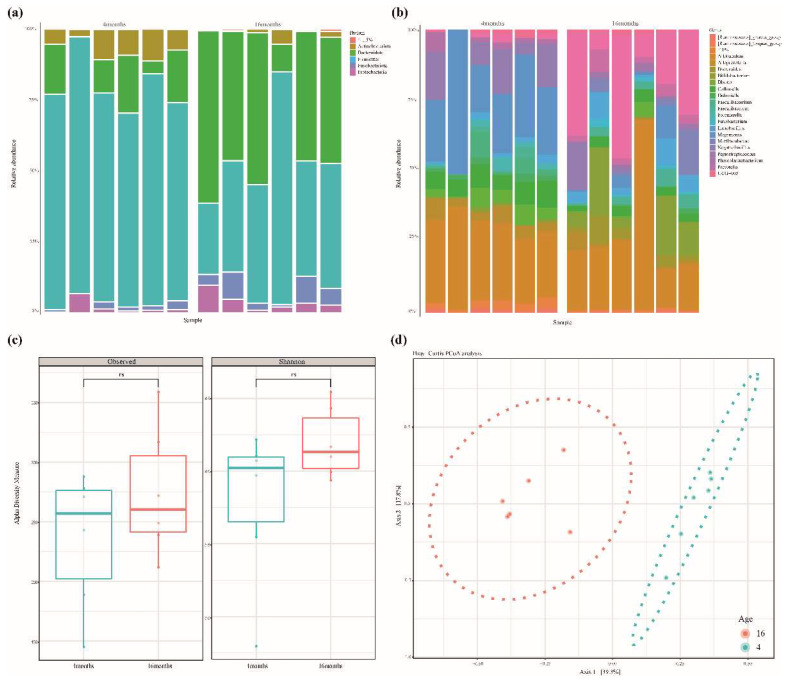
(**a**,**b**) Relative abundance of fecal microbiota at the phylum and genus level. (**c**) The alpha diversity of the fecal microbiome between the 4-month-old and 16-month-old groups. The richness of fecal microbiota was analyzed by the observed amplicon sequence variants (ASVs). Evenness was evaluated by the Shannon index. There were no significant differences in observed ASVs or Shannon value between growth stages (Kruskal–Wallis, observed-ASVs p-value = 0.42 and Shannon = 0.2). (**d**) Principal Coordination Analysis (PCoA) based on Bray–Curtis dissimilarity distance matrix. Beta diversity in the 4-month-olds (blue) and 16-month-olds (red) is grouped by bacterial compositional dissimilarities.

**Figure 2 animals-12-02499-f002:**
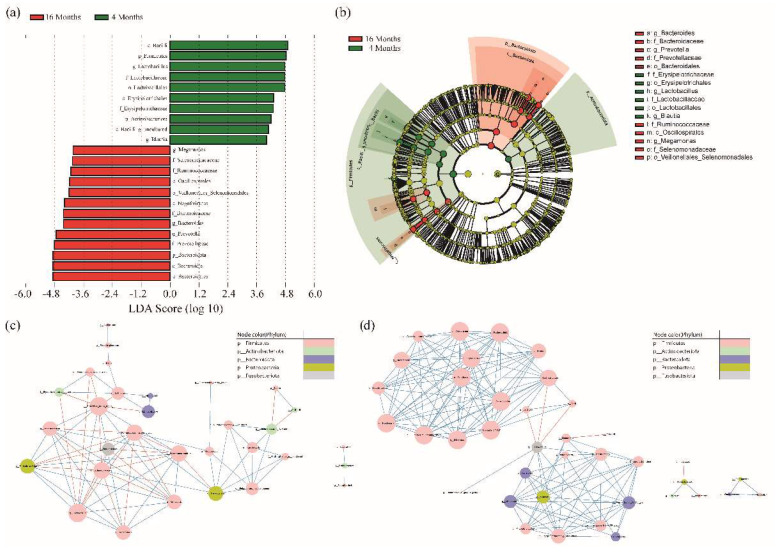
(**a**) To identify the significantly different abundant taxa between the 4 and 16−month−old groups, we used the LEfSe method. Statistically significant groups were reported with linear discriminant analysis (LDA) scores > 4. (**b**) The cladogram showed the taxonomic distribution of bacterial groups (green; 4−month−old, red; 16−month−old). (**c**,**d**) shows the Pearson correlation network of the gut microbiome genera. Each node represents a genus and the node size represents the number of related edge numbers. Blue and red edges indicate positive and negative associations between nodes, respectively. The node color indicates a phylum (coefficient value > |0.8|, shown in Appendix A).

**Figure 3 animals-12-02499-f003:**
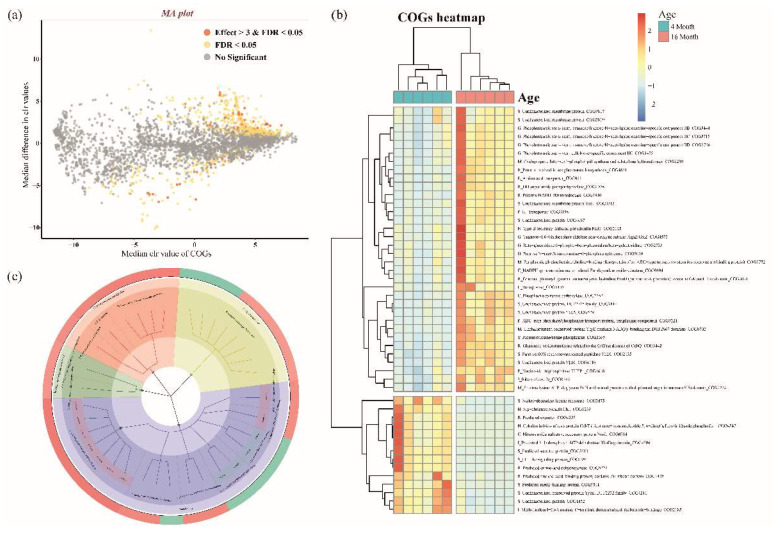
(**a**) Significantly distinct COGs determined by ALDEx2 algorithm between the 4 and 16-month-old group using the MA plot (BH adjusted *p*-value < 0.05 and effect size > 3 (distinct in 4 months) or effect size < −3 (distinct in 16 months)). Only the significantly distinct COGs are categorized by the function of COGs. (**b**) The heatmap presents the predicted functions based on significantly distinct COGs in the 4-month-old and 16-month-old groups. The heatmap colors indicate the normalized relative abundance of COGs numbers based on the KEGG pathway using the PICRUSt2. (**c**) Circular dendrogram of COGs of the 4-month-old and 16-month-old groups present the outliers, respectively, indicated in green and red. The second inner side, the second position from the most outside, showed COG classes indicating the “Cellular process and signaling” (red), “Information storage and processing” (emerald green), “Metabolism” (blue), and the “Poorly characterized” (yellow). The next inner side showed the COGs functional categories. “Metabolism” of COGs classes, “G, Carbohydrate transport and metabolism”, “E, amino acid transport and metabolism”, and “F, Nucleotide transport and metabolism” categories are shown in the 16-month-old group, and the “H, Coenzyme transport and metabolism” and “I, Lipid transport and metabolism” are shown in the 4-month-old group.

**Table 1 animals-12-02499-t001:** Summary of DADA2 sequence data quality control.

Age (Months)	Input Reads (Average)	Total Input Reads	Output Reads (Average)	Total Output Reads	Percentage of Input Non-Chimeric (Average)
4	93,915.3	563,492	52,662.83	315,977	56.03
16	87,963.5	527,781	49,767.17	298,603	56.55
Total	90,939.42	1,091,273	51,215	614,580	56.29

## Data Availability

The datasets generated for this study have been submitted to the NCBI database with reference to SRA submission SUB11980630.

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
