# Peer review of "Insight into the Fecal Microbiota Signature Associated with Growth Specificity in Korean Jindo Dogs Using 16S rRNA Sequencing"

_animals, 2022, doi:10.3390/ani12192499_

Round 1

Reviewer 1 Report

In general, this is a standard study on animals gut microbiota. The weak side of the study is a small amount of samples (6 animals).

Specific comments:

1. Raw sequencing data must be made publicly available. For example in NCBI as BioProject

2. Figure 1 as well as Figure 2 and 3 are practically unreadable. Very poor quality of figures.

3. It is not clear from the conclusions what results were obtained. Conclusions need to be written in more detail.

Reviewer 2 Report

Main comments

1. Where are these studied dogs housed in the owner’s home or animal facility? As we know, the environment of the animals living greatly impacts the intestinal microbiota community.

2. The intestinal microbiome has been studied in many human diseases by animal models, which can be further used for therapeutic and potential medical treatments. What the findings in this study can be used in the future was not clear, and it cannot convince me of the results with the very small cohort used in this study (just six dogs).

3. All the functional results were predicted based on the 16S rRNA sequencing data with the limited animal number, which weakened the novelty and significance of this study.

4. The quality of the figures was relatively low, and lots of the information could not be read.  

Minor comments

1. Line 55: References or citations should be provided here to support this claim.

2. Line 65-67: Please rewrite these sentences; the contents were repeated shown up.

Reviewer 3 Report

Thanks for your this work.

The only comment is why the author didn't support the results of the work by references related to the specific species of animal under investigation (dogs or even pets). Most of references are related to human beings.

Is the gut microbiome of human is similar to that of dog?

Best wishes
